# Enhanced Bioavailability and Efficacy of Silymarin Solid Dispersion in Rats with Acetaminophen-Induced Hepatotoxicity

**DOI:** 10.3390/pharmaceutics13050628

**Published:** 2021-04-28

**Authors:** Im-Sook Song, So-Jeong Nam, Ji-Hyeon Jeon, Soo-Jin Park, Min-Koo Choi

**Affiliations:** 1BK21 FOUR Community-Based Intelligent Novel Drug Discovery Education Unit, Vessel-Organ Interaction Research Center (VOICE), Research Institute of Pharmaceutical Sciences, College of Pharmacy, Kyungpook National University, Daegu 41566, Korea; goddns159@nate.com (S.-J.N.); kei7016@naver.com (J.-H.J.); 2College of Korean Medicine, Daegu Haany University, Daegu 38610, Korea; sjp124@dhu.ac.kr; 3College of Pharmacy, Dankook University, Cheon-an 31116, Korea

**Keywords:** silymarin, D-α-Tocopherol polyethylene glycol 1000 succinate (TPGS), liver distribution, acetaminophen-induced hepatotoxicity

## Abstract

We evaluated the bioavailability, liver distribution, and efficacy of silymarin-D-α-tocopherol polyethylene glycol 1000 succinate (TPGS) solid dispersion (silymarin-SD) in rats with acetaminophen-induced hepatotoxicity (APAP) compared with silymarin alone. The solubility of silybin, the major and active component of silymarin, in the silymarin-SD group increased 23-fold compared with the silymarin group. The absorptive permeability of silybin increased by 4.6-fold and its efflux ratio decreased from 5.5 to 0.6 in the presence of TPGS. The results suggested that TPGS functioned as a solubilizing agent and permeation enhancer by inhibiting efflux pump. Thus, silybin concentrations in plasma and liver were increased in the silymarin-SD group and liver distribution increased 3.4-fold after repeated oral administration of silymarin-SD (20 mg/kg as silybin) for five consecutive days compared with that of silymarin alone (20 mg/kg as silybin). Based on higher liver silybin concentrations in the silymarin-SD group, the therapeutic effects of silymarin-SD in hepatotoxic rats were evaluated and compared with silymarin administration only. Elevated alanine aminotransferase, aspartate aminotransferase, and alkaline phosphatase levels were significantly decreased by silymarin-SD, silymarin, and TPGS treatments, but these decreases were much higher in silymarin-SD animals than in those treated with silymarin or TPGS. In conclusion, silymarin-SD (20 mg/kg as silybin, three times per day for 5 days) exhibited hepatoprotective properties toward hepatotoxic rats and these properties were superior to silymarin alone, which may be attributed to increased solubility, enhanced intestinal permeability, and increased liver distribution of the silymarin-SD formulation.

## 1. Introduction

Milk thistle has been used for over 2000 years as a general medicinal herb to treat liver, kidney, and gallbladder diseases [1,2]. Silymarin is an ethanol extract from milk thistle and is a complex mixture of flavonolignans, consisting of silibin, isosilibin, silydianin, silychristin and other compounds [3]. Silybin, a main component of silymarin, accounts for about 60–70%, followed by silychristin (20%), silydianin (10%), and isosilybin (5%) [4,5]. Silymarin is one of the most popular herbal supplementations that are known to be effective in liver disease [6]. It is also used to protect liver toxicity induced by acute ethanol exposure, carbon tetrachloride treatment, and high acetaminophen (APAP) doses [7]. Hepatoprotective effects of silymarin are mediated by reducing reactive oxygen species and increasing cellular glutathione and superoxide dismutase levels in the liver [8].

Despite the therapeutic benefits of silymarin, it is used at high doses (280–1000 mg) due to its low aqueous solubility (50–430 μg/mL), low bioavailability (23–47%), and limited absorption properties [9,10,11]. These poor biochemical characteristics may lead to unsatisfactory and nonreproducible clinical outcomes, in spite of the high doses, as there is an increased possibility of drug–drug interactions with other concomitantly administered drugs [6,12,13]. Therefore, formulation strategies to increase silymarin solubility and intestinal absorption have been investigated [14]. Widespread approaches include lipid-based formulations, including emulsions, liposomes, and solid lipid nanoparticles. Silymarin-loaded emulation-containing soybean lecithin and Tween 80 resulted in a 1.9-fold increase in the oral bioavailability of silymarin compared with silymarin suspended in polyethylene glycol (PEG) [15]. A self-emulsifying drug delivery system (SEDDS) has been applied to silymarin formulation. Silymarin-loaded SEDDS containing Tween 20, HCO-50, and Transcutol increased the silymarin bioavailability 3.6-fold [16]. A 2.7-fold increase in silymarin oral bioavailability has also been reported by using silymarin SEDDS composed of ethyl linoleate, Cremophor EL, and ethanol [17]. However, these emulations and SEDDS formulations had a high content of surfactant or other excipients. Silymarin nanoemulsions with reduced surfactant content have been reported to enhance silymarin oral bioavailability ranged from 1.3- to 4-fold [18,19,20]. Liposomes, proliposomes, and PEGylated liposomes of silymarin composed of phospholipid and cholesterol have been reported. These formulations showed a high encapsulation efficiency of more than 85% with increasing oral bioavailability of silymarin [14,21,22]. Moreover, surface-modified liposomes were reported to increase hepatoprotective efficacy. Hepatic-targeting ligand Sito-G-modified PEGylated silymarin liposomes were formulated and they enhanced hepatic uptake of silymarin in HeG2 cells [23]. Solid lipid nanoparticles of silymarin using Compritol 888 ATO, soybean lecithin, and poloxamer 188 showed a 2.8-fold higher oral bioavailability and enhanced liver distribution [24]. Besides the increased solubility and bioavailability of silymarin, in vivo liver-targeting and therapeutic efficiency await further investigation. Many other silymarin formulations, such as cyclodextrin inclusion complexes, solid dispersions (SDs), polymer-based nanocarriers, and so on [14], have focused on the linkage between the solubility enhancement and their oral bioavailability because the low silymarin solubility has been reported to limit its absorption [14,25].

In addition to low solubility, silymarin acts as a substrate for efflux pumps in the intestinal epithelium, and silymarin absorption is therefore restricted by these mechanisms, such as P-glycoprotein (P-gp), breast cancer resistance protein (BCRP), and multidrug resistance-related proteins (MRPs) [26]. Because of the nature of silymarin (e.g., low solubility and low permeability), it is classified under the Biopharmaceutics Classification System (BCS) as Category IV [14,27]. However, there have been controversies regarding the BCS category of silymarin. Many research papers consider it as BCS Category II [28,29]. Therefore, the objective of this study was to increase the oral bioavailability of silymarin by increasing its intestinal permeability through the inhibition of an efflux transporter, which is a strategy for the formulation of BCS Category III or IV drugs showing low intestinal permeability [30,31,32]. SD formulation was selected because it is a well-established technique for increasing the oral bioavailability by increasing the solubility of poorly-soluble drugs and by increasing permeability by modulating intestinal efflux pumps. Moreover, SD is easy to formulate with a reduced drug-to-excipient ratio and it is easy to change the kind of excipient [33].

Recently, various pharmaceutical excipients have emerged, not only as solubilizing agents, but also as potential alternatives to P-gp and metabolic inhibitors [32]. Polyethylene glycol 400 (PEG400), pluronic P85, pluronic F127, Tween 80, and vitamin E-D-α-tocopheryl polyethylene glycol 1000 succinate (TPGS) have been reported to inhibit in vitro P-gp-mediated efflux and intestinal metabolism when assessed using digoxin and verapamil as a substrate for P-gp and cytochrome P450 (CYP) 3A, resulting in enhanced oral bioavailability [30,31,32,34]. Among these excipients, TPGS is a nonionic water soluble vitamin E derivative, approved for pharmaceutical adjuvant use in drug formulations by the United States Food and Drug Administration [35]. TPGS has been used as an absorption enhancer, emulsifier, solubilizing agent, and permeation enhancer in pharmaceutics and cosmetics [36,37,38]. The molecule prolongs the half-life of drugs, and increases intestinal drug absorption by inhibiting ATPase activity of efflux pumps [30,39,40].

Therefore, we aimed to investigate the solubility and intestinal permeability of a silymarin-TPGS solid dispersion (silymarin-SD) formulation. We also aimed to investigate the pharmacokinetics and liver distribution, the target tissue of silymarin, as well as the efficacy of our silymarin-SD formulation in rats with acute hepatotoxicity. Concentrations of silymarin were monitored as silybin, the representative and major component of silymarin [3,41]. To induce acute hepatotoxicity, we used APAP overdosing, which is a widely used chemically-induced hepatoxic model [42]. At safe therapeutic doses, APAP is metabolized into glucuronide and sulfate via conjugation reactions at 60–90%. Approximately 5–10% of APAP is oxidized to N-acetyl-p-benzoquinone imine (NAPQI) by mixed-function oxidase enzymes. APAP is immediately conjugated with glutathione [16,17,18,19,20,21]. However, APAP overdoses produce the reactive intermediate, NAPQI [43], which binds to cellular proteins, induces oxidative stress, and promotes injury development during APAP-induced hepatotoxicity [30,44].

## 2. Materials and Methods

### 2.1. Reagents

Silymarin flabolignans (CAS No. 65666-07-1), Silybin (CAS No. 22888-70-6), TPGS (CAS No. 9002-96-4), Poloxamer 407 (CAS No. 9003-11-6), Tween 20 (CAS No. 9005-64-5), Tween 80 (CAS No. 9005-65-6), hydroxypropyl cellulose (HPC; CAS No. 9004-64-2), sodium dodecyl sulfate (SDS; CAS No. 151-21-3), pluronic F127 (CAS No. 9003-11-6), polyvinylpyrrolidone (PVP; CAS No. 9003-39-8), polyethylene glycol 400 (PEG 400) (CAS No. 25322-68-3), sodium carboxymethyl cellulose (CMC) (CAS No. 9004-32-4), naringenin (internal standard, IS) (CAS No. 67604-48-2), Hank’s balanced salt solution (HBSS, pH 7.4), and acetaminophen (APAP; CAS No. 103-90-2) were purchased from Sigma-Aldrich (St. Louis, MO, USA). Distilled water, acetonitrile, and methanol were of high-performance liquid chromatography grade and purchased from J.T. Baker (Center Valley, PA, USA). All other chemicals and solvents were of reagent and analytical grade.

### 2.2. Selection of Excipients

The effect of various excipients on the silymarin solubility was measured. Silymarin (5 mg) and excipients such as TPGS, Poloxamer 407, Tween 20, Tween 80, HPC, SDS, pluronic F127, PVP (5 mg each) were sonicated for 5 min with 2 mL of distilled water and shaken at 25 °C for 12 h. Mixtures were centrifuged at 16,000× *g* for 10 min and supernatants filtered through a membrane filter (pore size; 0.22 μm). Filtrates were diluted in 85% acetonitrile and silybin concentrations analyzed using a liquid chromatography-tandem mass spectrometry (LC-MS/MS) system.

### 2.3. Preparation of Silymarin-SD

To determine the ratio of silymarin and TPGS, the solubility of silymarin (10 mg) was measured in the presence of various concentrations of TPGS (0.01–50 mg). After dissolving silymarin (10 mg) and varying concentrations of TPGS (0.01–50 mg) in 30 mL of 40% ethanol and freezing at −80 °C for 12 h, this mixture was then dried using a freeze dryer (FDCF-12012, operon, Seoul, Korea) at −120 °C for 72 h. The solubility of the silymarin formulation with varying ratios of TPGS was measured after dissolving these formulations in 2 mL of distilled water and shaken at 25 °C for 12 h. Mixtures were centrifuged at 16,000× *g* for 10 min and supernatants filtered through a membrane filter (pore size; 0.22 μm). Filtrates were diluted in 85% acetonitrile and silybin concentrations were analyzed using a liquid chromatography-tandem mass spectrometry (LC-MS/MS) system.

After deciding the ratio of silymarin and TPGS, SD formulation of silymarin-TPGS at a ratio of 1:1 (*w*/*w*) (silymarin-SD) was prepared for further characterization. Briefly, silymarin and TPGS were accurately weighed (0.5 g each), and dissolved in 300 mL of 40% ethanol and frozen at −80 °C for 12 h and this mixture was then dried using a freeze dryer (FDCF-12012, operon, Seoul, Korea) at −120 °C for 72 h. After freeze-drying, the resulting samples were passed through a KP sieve (mesh size = 0.84 μm) and stored in a thermo-hygrostat (25 °C, 20% relative humidity) until the use of silymarin-SD for the characterization and the pharmacokinetic and efficacy study.

### 2.4. Characterization of Silymarin-SD

Silymarin (5 mg) and silymarin-SD (10 mg) was dissolved with 2 mL of distilled water and shaken at 25 °C for 12 h. Mixtures were centrifuged at 16,000× *g* for 10 min and supernatants filtered through a membrane filter (pore size; 0.22 μm). Filtrates were diluted in 85% acetonitrile and silybin concentrations analyzed using an LC-MS/MS system.

Dissolution studies were conducted in 900 mL of distilled water for 120 min in a D-63150 dissolution test apparatus (Erweka, Heusenstamm, Germany) at 37 °C and 50 rpm using a paddle method. Briefly, silymarin powder (20 mg) and silymarin-SD formulation (equivalent to 20 mg silymarin) were packed into a gelatin capsule, and each capsule was placed inside the sinker. An aliquot (1 mL) of a medium was collected at 0, 10, 20, 30, 45, 60, 90, and 120 min and filtered through a membrane filter (pore size; 0.22 μm), and an equal volume of water replaced after each sampling. Silybin concentrations in filtrates were analyzed using an LC-MS/MS system.

X-ray diffraction (XRD) of silymarin, TPGS, and silymarin-SD was determined on an X-ray diffractometer (Ultima IV; Rigaku Co., Tokyo, Japan) using Cu Kα radiation, at 40 mA and 40 kV. Data were obtained from 5–60° (2θ) at a step size of 0.02° and a scanning speed of 5°/min.

Differential scanning calorimetry (DSC) of silymarin, TPGS, and silymarin-SD was determined using a DSC 131EVO (Setaram, Caluire, France). Sample weighing approximately 5 mg were placed in a closed aluminum pan and heated at a scanning rate of 5 °C/min from 10 °C to 200 °C, with nitrogen purging at 20 mL/min. Indium was used to calibrate the temperature scale.

Fourier-transform infrared spectroscopy (FTIR) spectra of silymarin, TPGS, physical mixture, and silymarin-SD were obtained in the spectral region of 4000–600 cm^−1^ using a resolution of 4 cm^−1^ and 64 scans using a Frontier FTIR spectrometer (PerkinElmer, Norwalk, CT, USA) in transmittance mode.

### 2.5. Intestinal Permeability Study

Male Wistar rats weighing 225–270 g (eight weeks old, *n* = 28) were purchased from Samtako Co. (Osan, Korea). Rats were housed in a 12-h light/dark cycle, and food and water were supplied ad libitum for one week prior to animal studies. Control rats (*n* = 4) received a vehicle (40% PEG 400) by oral administration for two days (4 p.m., 11 p.m., and 9 a.m.). APAP rats (*n* = 4) received an oral dose of APAP (3 g/kg dissolved in 40% PEG 400) for two days (4 p.m., 11 p.m., and 9 a.m.). Rats were fasted for 16 h, but had free access to water before study commencement. Rats were then anesthetized using isoflurane (isoflurane vaporizer to 2% with oxygen flow at 0.8 L/min) 24 h after the last APAP administration. Blood samples were collected from the abdominal aorta in heparinized blood tubes, and centrifuged at 16,000× *g* for 1 min to separate plasma. These were used to determine alanine aminotransferase (ALT) and aspartate aminotransferase (AST) levels using kits supplied by Young Dong Diagnostics (Yongin, Korea). A proximal jejunum section (approximately 10 cm) was excised and washed in prewarmed HBSS (pH 7.4). Segments were mounted on a tissue holder of a Navicyte Easy Mount Ussing Chamber (Warner Instruments, Holliston, MA, USA), with a surface area of 0.76 cm², and acclimated in HBSS for 15 min with continuous oxygenation (95% O₂/5% CO₂ gas). Intestinal permeability studies were commenced by changing HBSS on both sides of intestinal segments using 1 mL prewarmed HBSS containing silymarin or silymarin-SD (i.e., equivalent to 20 μM silybin) at the donor side and 1 mL prewarmed fresh HBSS at the receiver side. Sample aliquots (400 μL) were withdrawn every 30 min for 2 h from the receiver side, and an equal volume of prewarmed fresh HBSS was replaced at the receiver side.

To investigate whether efflux transporters were involved in this process, an apical to basal (A to B) and a basal to apical (B to A) transport of 20 μM silybin in the presence of representative inhibitors of P-gp, MRPs, and BCRPs such as 20 μM cyclosporine A (CsA), 100 μM MK-571, and 20 μM fumitremorgin C (FTC), respectively, were measured in a proximal jejunum section from control rats (*n* = 4) [45,46]. The effects of TPGS (0.01–1 mg/mL) on silybin permeability were also measured in a proximal jejunum section from control rats (*n* = 4) using the above method. For silybin analysis, sample aliquots (100 μL) were mixed for 5 min with 200 μL acetonitrile containing 20 ng/mL of naringenin (IS), and centrifuged at 16,000× *g* for 5 min. After this, a 5 μL supernatant was directly injected into an LC-MS/MS system.

To monitor intestinal integrity, Lucifer yellow (50 μM) permeability was measured in a proximal jejunum section from control rats (*n* = 12) as described previously [31,45]. Lucifer yellow fluorescence in 200 μL sample aliquots was measured using a fluorescence spectrophotometer (Infinite 200 PRO, Tecan, Switzerland) at an excitation wavelength of 425 nm and emission wavelength of 535 nm.

### 2.6. Pharmacokinetics of Silymarin or Silymarin-SD in APAP-Induced Hepatotoxic Rats

Pharmacokinetics of silymarin following single and repeated oral administration of silymarin and silymarin-SD for 5 days was measured in APAP-induced hepatotoxic rats. To induce hepatotoxicity, rats received an oral dose of APAP (3 g/kg) for 2 days, which was identical to the method described in Section 2.4. The dosing schedule for silymarin or silymarin-SD and APAP is shown in Figure 1.

For single administration of silymarin or silymarin-SD, rats with APAP-induced hepatotoxicity received silymarin (20 mg/kg as silybin, *n* = 4) or silymarin-SD (20 mg/kg as silybin, suspended in 0.5% CMC, *n* = 4) via oral gavage 24 h after the last APAP administration. Blood samples were taken from the cannulated femoral artery at 0.25, 0.5, 1, 2, 4, 8, and 24 h after silymarin and silymarin-SD administration (Figure 1). Bloods were centrifuged at 16,000× *g* for 1 min, and 50 μL plasma stored at −80 °C for silybin assay.

For the repeated administration of silymarin and silymarin-SD for five consecutive days, rats received silymarin (20 mg/kg as silybin, *n* = 4) or silymarin-SD (20 mg/kg as silybin, suspended in 0.5% CMC, *n* = 4) via oral gavage at 10 a.m. or five consecutive days. On the third and fourth day, rats received an oral APAP (3 g/kg dissolved in 40% PEG 400) dose at 4 p.m., 11 p.m., and 9 a.m.. Blood samples were taken from the cannulated femoral artery at 0.25, 0.5, 1, 2, 4, 8, and 24 h after the last administration of silymarin or silymarin-SD (Figure 1), centrifuged at 16,000× *g* for 1 min, and 50 μL plasma were stored at −80 °C for the silybin assay.

To investigate the liver distribution of silymarin and silymarin-SD, blood samples were collected from the abdominal artery at 0.5, 1, 2, 4, and 24 h after the repeated oral administration of silymarin (20 mg/kg as silybin, *n* = 20) or silymarin-SD (20 mg/kg as silybin, *n* = 20), according to the dosing schedule. Immediately after blood sampling, the liver was excised, rinsed in saline, and homogenized in nine volumes of saline, using a tissue homogenizer. Aliquots (50 μL) of 10% liver homogenate samples were stored at −80 °C for silybin assay.

Silybin concentrations in plasma and 10% liver homogenate samples were analyzed. Samples (100 μL) were added to 300 μL IS (naringenin, 20 ng/mL in acetonitrile) and the mixture was vigorously mixed for 10 min, followed by centrifugation at 16,000× *g* for 5 min. Supernatant aliquots (5 μL) were directly injected into an LC-MS/MS system.

### 2.7. LC-MS/MS Analysis of Silybin

The analysis of the silybin concentration was performed using an Agilent 6430 triple quadrupole LC/MS-MS system (Agilent, Wilmington, DE, USA) coupled to an Agilent 1290 HPLC system according to the previous method with slight modification [41]. Silybin and naringenin (IS) was separated on Synergi Polar RP column (150 × 2 mm, 5 μm particle size, Phenomenex, Torrance, CA, USA) with a mobile phase consisting of acetonitrile containing 0.1% formic acid: distilled water containing 0.1% formic acid = 85:15 (*v*/*v*) at a flow rate of 0.2 mL/min. Column oven temperature was maintained at 30 °C.

Multiple reactions monitoring conditions for silybin and naringenin (IS) in a negative ionization mode were used at *m*/*z* 481.1 → 301.0 for silybin and *m*/*z* 271.1 → 151.3 for naringenin with a collision energy of 15 eV. Quantitation was performed using the Agilent Mass Hunter Qualitative Analysis B. 04.00 software. A standard curve for silybin (2–500 ng/mL) was prepared by serially diluting silybin stock solution. The intra-day and inter-day precision and accuracy variations were within 15%.

### 2.8. Efficacy of Silymarin or Silymarin-SD in APAP-Induced Hepatotoxic Rats

Silymarin or silymarin-SD efficacy in APAP-induced hepatotoxic rats following repeated oral administration of silymarin or silymarin-SD for five days was measured using the same dosing schedule (Figure 1). Rats received vehicle (0.5% CMC, *n* = 4), TPGS (50 mg/kg suspended in 0.5% CMC, *n* = 4), silymarin (20 mg/kg as silybin suspended in 0.5% CMC, *n* = 4), and silymarin-SD (20 mg/kg as silybin, suspended in 0.5% CMC, *n* = 4) via oral gavage at 10 a.m. for five consecutive days. On the third and fourth day, rats received an oral APAP dose (3 g/kg dissolved in 40% PEG 400) at 4 p.m., 11 p.m., and 9 a.m. Twenty-four hours (24 h) after the last administration of vehicle, silymarin, TPGS, and silymarin-SD, blood samples were taken from the abdominal artery and centrifuged at 16,000× *g* for 1 min to collect plasma. Sample aliquots (200 μL each) were used to quantify markers of liver function, including ALT, AST, alkaline phosphatase (ALP), total cholesterol (TC), high-density lipoprotein cholesterol (HDL-C), and low-density lipoprotein cholesterol (LDL-C). Levels were measured at Seoul Clinical Laboratories (Yongin, Korea).

Immediately after blood sampling, liver tissues were excised from all rats and fixed in formalin buffer solution. Liver sections of 3 μm thick were sectioned and stained in hematoxylin and eosin (H&E). Histopathological observations were conducted by the Korea Pathology Technical Center (Cheongju, Korea).

### 2.9. Data Analysis

Pharmacokinetic parameters were calculated using the WinNonlin 5.1 using non-compartmental analysis. The data are expressed as the means ± standard deviation for the groups. Statistical analysis was performed using the Student t-test and one-way analysis of variance (ANOVA) test.

## 3. Results

### 3.1. Preparation of Silymarin-SD

The effect of various excipients, which have been proved for the solubility enhancement of silymarin and for the modulation of efflux transporters in the literature [29,32,33,34], on the silymarin solubility was measured with a silymarin to excipient ratio of 1:1 (*w*/*w*). Among the tested excipients, SDS, Tween 20, and HPC increased silymarin solubility by less than 1.5-fold. On the other hand, poloxamer 407, Tween 80, and pluronic F127 increased the silymarin solubility by 2~3-fold. Addition of PVP and TPGS showed the highest increase in silymarin solubility (Figure 2A). Finally, TPGS was selected as an excipient for silymarin formulation based on the previous results that showed TPGS inhibits the efflux transporters including P-gp [30,39,40].

To decide the ratio between silymarin and TPGS, SD of silymarin-TPGS with varying ratios of silymarin and TPGS (i.e., 1:0.001–1:5 *w*/*w*) was prepared by the freeze-drying method. Silymarin solubility, which was monitored by silybin concentration, a representative component of silymarin [41], increased sharply by the addition of TPGS up to 10 mg, and increased steadily when adding up to 50 mg TPGS (Figure 2B). Therefore, the silymarin-to-TPGS ratio was set at 1:1 for the preparation of silymarin-TPGS solid dispersion (silymarin-SD) formulations. After preparing silymarin-SD, the loading efficiency of silymarin in silymarin-SD was 50.9%.

### 3.2. Characterization of Silymarin-SD

When silymarin-SD solubility was compared with silymarin alone, a 23-fold increase in silybin solubility was observed compared with that of silymarin itself (Figure 3A). Not only was this solubility increased, but the dissolution rate of silymarin-SD was greater than silymarin alone (Figure 3B).

Silymarin, TPGS, and silymarin-SD XRD patterns are shown in Figure 3C. Silymarin and TPGS exhibited sharp peaks in a 2θ angle ranging from 10 to 30, indicating a typical crystalline structure for both. The diffraction peaks for silymarin-SD decreased markedly when compared with silymarin and TPGS, suggesting ingredients in the solid dispersion were in an amorphous state. Enhancement of silymarin solubility was 3.8-fold when the silymarin solubility of silymarin-SD prepared by the freeze-drying method was compared with that in the combined silymarin and TPGS at the same ratio (Figure 2A,B), suggesting that the 3.8-fold increase in the silymarin solubility can be explained by formulating silymarin-SD as amorphous state.

The DSC results are shown in Figure 3D. Silymarin produced a wide endothermic peak at about 90 °C and glass transition temperature of 70 °C, which was consistent with previous results [33] and indicated its crystalline nature. TPGS produced a sharp endothermic peak at 33.75 °C and glass transition temperature of 30.7 °C, which was also consistent with previous results [30]. However, in the DSC thermogram of silymarin-SD, the indicative peaks for silymarin and TPGS disappeared with a slight decrease around 75 °C and an increase around 150 °C. It could be concluded that silymarin and TPGS in the silymarin-SD were in the amorphous form after the fabrication.

The FTIR pattern of silymarin, TPGS, physical mixture (PM) of silymarin and TPGS, and silymarin-SD are shown in Figure 3E. The FTIR pattern of individual silymarin and TPGS was similar to previous results [29,33]. The FTIR spectrum of PM exhibited characteristic peaks of both silymarin and TPGS and showed similar peaks to the silymarin-SD, suggesting that there was no major shift of peaks and no existence of covalent interaction between silymarin and TPGS.

### 3.3. Enhanced Intestinal Permeability of Silymarin-SD

To investigate transporter efflux in silymarin transport in rat intestines, we investigated the permeability of silybin, a representative component of silymarin [41], in the presence of P-gp, BCRP, and MRP inhibitors (Figure 4C). Efflux ratios, calculated by dividing B to A permeability (P_app,BA_) by A to B permeability (P_app,AB_) [47], of silybin was 5.8 and decreased to 1.7 and 0.7 by the presence of CsA (P-gp inhibitor) and MK571 (MRP2 inhibitor), respectively. However, FTC (BCRP inhibitor) treatment did not alter the efflux ratio of silybin (i.e., 5.5). In addition, TPGS also decreased the P_app,BA_ of silybin and increased the P_app,AB_ of silybin in a TPGS concentration-dependent manner (Figure 4A). In addition, efflux pump inhibitors did not alter the P_app_ of Lucifer yellow, a marker of cell integrity [45], whereas TPGS significantly increased the P_app_ of Lucifer yellow (Figure 4B). These results suggested that P-gp and MRPs were involved in silybin efflux, and TPGS increased silybin absorption by inhibiting silybin efflux. TPGS also functioned as a permeation enhancer to increase the paracellular pathway of silybin by disturbing cell integrity.

Next, we measured the P_app_ of silymarin and silymarin-SD in control and APAP-induced hepatotoxic rats. We observed no significant differences in Papp values between the silymarin and silymarin-SD groups (Figure 4C,D). Moreover, the P_app,AB_ of Lucifer yellow in the intestinal segment of APAP-induced hepatotoxic rats was 2.8 × 10^−7^ cm/s, similar to control rats (Figure 4E, gray bar). This observation suggested that APAP treatment did not harm cell integrity or alter intestinal permeability [45].

However, P_app,BA_ of silybin was 4.5- and 5.8-fold greater than P_app,AB_ in both control and APAP rats, suggesting that the efflux pump was involved in intestinal absorption processing of silymarin; this can be a limiting factor for silymarin absorption [32]. When we compared the P_app_ of silybin in the silymarin and silymarin-SD groups (Figure 4C,D), P_app,AB_ of silybin increased as silymarin-SD by 3.2- and 4.6-fold compared with silymarin itself in the control and APAP groups, respectively. P_app,BA_ of silybin was decreased by the presence of silymarin-SD by 0.5- and 0.4-fold compared with silymarin itself in the control and APAP groups, respectively. The P_app,AB_ of Lucifer yellow was not different in the presence of silymarin itself but significantly increased with the addition of silymarin-SD (Figure 4E). Thus, by using silymarin-SD, the efflux ratio of silybin was decreased 0.6-fold and the P_app,AB_ of silybin was increased 4.6-fold, through the inhibition of efflux of silybin, and the enhancement of paracellular permeability.

### 3.4. Increased Bioavailability of Silymarin-SD

Plasma concentration–time silybin profiles after single and repeated oral administration of silymarin and silymarin-SD are shown (Figure 5). Pharmacokinetic parameters, as calculated from plasma concentration–time profiles, are presented in Table 1. In the single oral administration group, the maximum plasma concentration (C_max_) and the area under the plasma concentration curve (AUC) for silybin following silymarin-SD administration were 4.0- and 1.6-fold, respectively, greater than those for silymarin only. However, the time to reach C_max_ (T_max_) and elimination half-life (T_1/2_) was not different between the silymarin and silymarin-SD groups after the single oral administration (Figure 5A and Table 1). These results suggested that increased silybin plasma concentrations in the silymarin-SD group were caused by enhanced absorption.

We observed a significant increase in C_max_ and AUC of silybin after repeated oral administration of silymarin-SD for five consecutive days when compared with the repeated silymarin treatment group (Figure 5B and Table 1). However, the fold increase in the silybin AUC was greater (3.7-fold) in the repeated administration of the silymarin-SD group than silymarin itself (1.6-fold). These results suggested that the repeated silymarin-SD treatment may have increased the mean residence time (MRT) of silybin and this results in the accumulated plasma concentrations of silybin when administered repeatedly as silymarin-SD formulation.

Next, we investigated the liver distribution of silymarin-SD when compared with silymarin itself after repeated oral administration of silymarin-SD. Liver distribution is critical for the therapeutic efficacy of silymarin-SD because hepatoprotective effects of silymarin are mediated by reducing reactive oxygen species and increasing cellular glutathione and superoxide dismutase levels in the liver [8]. The liver concentrations of silybin in the silymarin-SD treatment group were significantly higher than those in the silymarin treatment group (Figure 6A). In addition, the liver silybin concentration in the silymarin group decreased sharply for 4 h, but was maintained for 24 h in the silymarin-SD group (Figure 6A). Therefore, the liver to plasma AUC ratio of silybin was significantly higher in the silymarin-SD than the silymarin group (Figure 6B).

### 3.5. Effect of Silymarin-SD on the APAP-Induced Hepatotoxicity

Based on higher liver silymarin concentrations of silymarin-SD compared with silymarin only after repeated oral administration, we evaluated the therapeutic effects of silymarin-SD in study rats. As shown in Figure 7, the biochemical hepatotoxicity markers, ALT, AST, and ALP were significantly increased by APAP treatment. These were decreased by the treatment with TPGS and silymarin only (APAP + TPGS and APAP + silymarin groups). However, decreased ALT, AST, and ALP levels were greater in the silymarin-SD treatment group compared with the TPGS and silymarin groups. Alterations in TC levels were not as significant as for ALT and AST, but TPGS, silymarin, and silymarin-SD treatments decreased these levels. When HDL-C and LDL-C levels were examined, silymarin-SD treatment was effective for APAP-induced hepatotoxicity, and more effective than the TPGS and silymarin groups.

Histopathology images from liver sections near the central vein are shown in Figure 8. APAP-toxicity sections showed inflammatory cell infiltration and disarranged hepatic cells (Figure 8B) compared to controls (Figure 8A). TPGS, silymarin, and silymarin-SD treatment groups showed less inflammatory cell infiltration compared with the APAP group (Figure 8C–E). When biochemical and histological results were combined, silymarin-SD treatment indicated superior protective effects toward liver tissue when compared with the TPGS only or silymarin only groups.

## 4. Discussion

Herbal based-dietary supplements are increasingly popular with more than 60% of adults taking these herbal supplements [48]. Silymarin, the concentrated extract of milk thistle, is one of the six best-selling herbal supplements and has long been used as a treatment for liver disease [12]. Despite the popularity of silymarin and its poor chemical properties and limited bioavailability, studies have attempted to enhance this bioavailability [37], which is limited by low solubility and efflux pump-mediated low intestinal permeability [13,26]. In recent years, several pharmaceutical excipients have emerged, not only as solubilizing agents, but also as potential inhibitors of intestinal first pass effect [32]. TPGS has been used as a solubilizer, stabilizer, permeation enhancer, and absorption enhancer, and P-gp inhibitors in a wide range of nanoliposomes, emulsions, micelles, and solid dispersions [30,38].

By formulating amorphous solid dispersions with silymarin and TPGS using freeze-drying methods, silybin solubility was increased 23-fold and intestinal permeability increased 4.6-fold (Figure 2 and Figure 4). In addition, the efflux ratio of silybin was decreased from 5.8 to 0.6 (Figure 4). Considering the efflux ratio of silybin was decreased by the presence of P-gp and MRP2 inhibitors and Lucifer yellow permeability was also enhanced by the presence of silymarin-SD, the increased intestinal permeability of silybin could be attributed to the permeation-enhancing effect and inhibition of P-gp- and MRP2-mediated silybin efflux. This enhanced permeation and dissolution rate of silybin in a silymarin-SD formulation generated increased silybin plasma concentrations. The AUC of silybin was increased 1.6-fold by the single oral administration of silymarin-SD when compared with silymarin administration (Figure 5A, Table 1). From the repeated administration of silymarin-SD for five days, plasma AUC was increased 2.1-fold when compared to the single administration of silymarin-SD and increased 3.7-fold when compared to the repeated administrations of silymarin itself (Figure 5B, Table 1). Moreover, silybin liver distribution was increased 3.3-fold when compared to the repeated administration of silymarin alone (Figure 6). Increased silybin accumulation by repeated administration of silymarin-SD may be attributed to decreased silybin elimination, consistent with increased MRT of silybin in the silymarin-SD group.

This increased liver distribution may result in enhanced therapeutic efficacy of si-lymarin-SD. The pre-treatment and co-treatment of silymarin-SD during APAP-induced hepatotoxicity showed superior effects in reducing hepatotoxic biomarkers, such as ALT, AST, and ALP compared with pre-treatment and co-treatment with silymarin itself. Since TPGS also reduces oxidative stress [38], we investigated the effects of TPGS on APAP-induced hepatotoxicity. As shown in Figure 7, TPGS exhibited partial activity on APAP-induced hepatotoxicity. Therefore, the superior effects of silymarin-SD increased liver concentrations of silybin and the resultant hepatoprotective effect could be attributed to the partial effect of TPGS. Taken together, the solid dispersion formulation of silymarin with TPGS not only enhanced bioavailability and increased distribution to target tissues, but also reinforced the practicality of functional excipient like TPGS (i.e., solubilizing effects, permeation enhancer, efflux pump modulation, and antioxidative effects) [38,40,49].

In conclusion, the solid dispersion formulation of silymarin-TPGS may be used to increase solubility and intestinal permeability; therefore, multiple silymarin-SD treatment (at a dose of 20 mg/kg as silybin) exhibited higher plasma concentrations and better hepatoprotective properties than the treatment of silymarin alone in APAP-induced hepatotoxic rats.

## Figures and Tables

**Figure 1 pharmaceutics-13-00628-f001:**
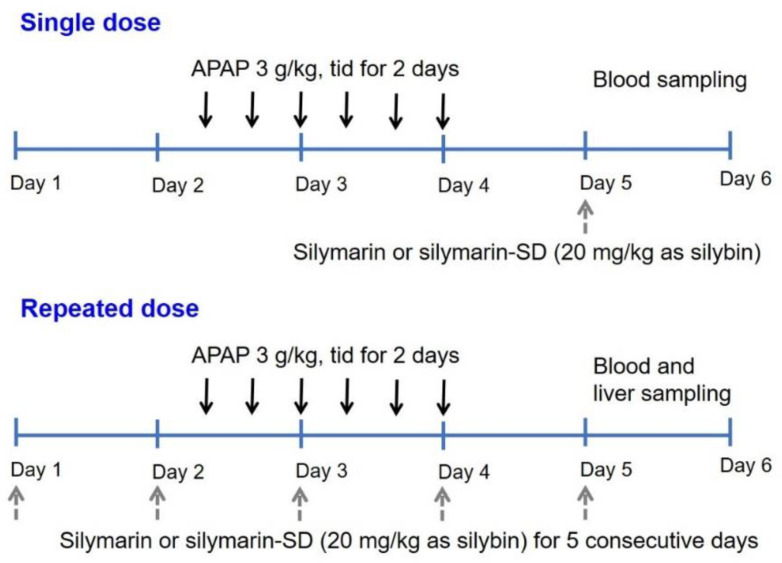
Dosing schedule for silymarin or silymarin-SD and APAP. Tid, three times per day.

**Figure 2 pharmaceutics-13-00628-f002:**
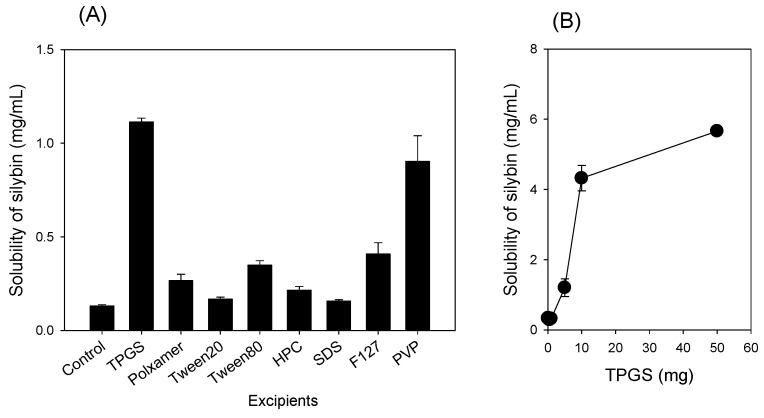
(**A**) Solubility of silymarin, which was monitored by silybin concentration, was measured in the presence of TPGS, Poloxamer407 (Poloxamer), Tween 20, Tween 80, hydroxypropyl cellulose (HPC), sodium dodecyl sulfate (SDS), pluronic F127 (F127), and polyvinylpyrrolidone (PVP). (**B**) Solubility of silymarin in the presence of varying amount of silymarin after preparing solid dispersion by freeze drying method. Each data represents mean ± standard deviation of triplicated determination.

**Figure 3 pharmaceutics-13-00628-f003:**
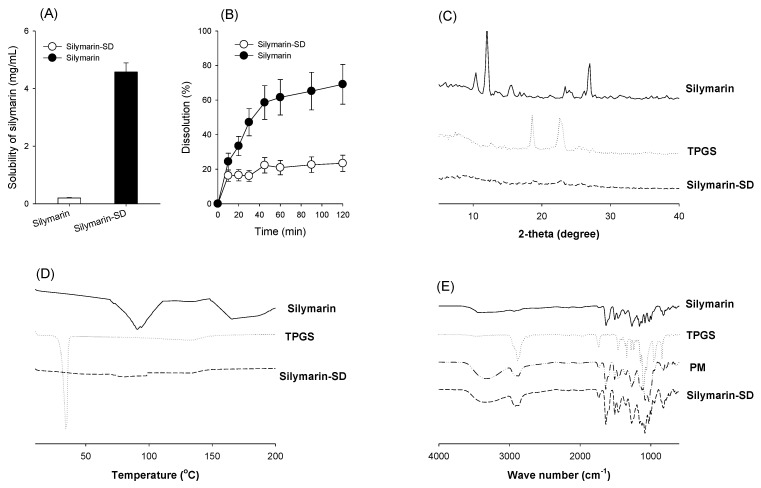
(**A**) Solubility and (**B**) dissolution rate of silymarin and silymarin-SD, which was monitored by silybin concentration, was measured. Each data represents mean ± standard deviation of triplicated determination. (**C**) X-ray diffraction (XRD) patterns of silymarin, TPGS, and silymarin-SD. (**D**) Differential scanning calorimetry (DSC) thermogram of silymarin, TPGS, and silymarin-SD. (**E**) Fourier-transform infrared spectroscopy (FTIR) spectrometer of silymarin, TPGS, a physical mixture of silymarin and TPGS (PM), and silymarin-SD.

**Figure 4 pharmaceutics-13-00628-f004:**
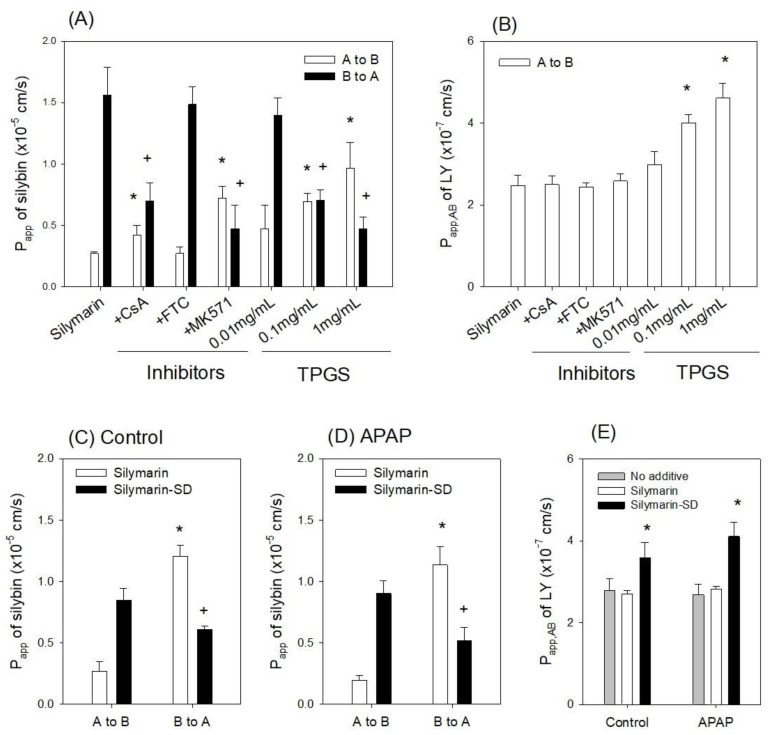
(**A**) The effect of efflux pump inhibitors and TPGS (0.01 mg/mL–1 mg/mL) on silybin permeability (P_app_) in rat jejunal segments. Twenty micromole (20 μM) cyclosporine A (CsA), 20 μM fumitremorgin C (FTC), and 100 μM MK-571 were used as inhibitors of P-gp, BCRP, and MRPs, respectively. (**B**) The effects of efflux pump inhibitors and TPGS (0.01 mg/mL–1 mg/mL) on A to B permeability (P_app,AB_) of Lucifer yellow (LY) in rat jejunal segments. The P_app_ of silybin in the presence of silymarin and silymarin-SD in jejunal segments from (**C**) control and (**D**) APAP-induced hepatotoxic rats. (**E**) The P_app_ of LY in the presence of silymarin and silymarin-SD in jejunal segments of control and APAP-induced hepatotoxic rats. Each data point represents the mean ± standard deviation (*n* = 4). * *p* < 0.05 compared with A to B P_app_ of silybin or LY. ^+^
*p* < 0.05 compared with B to A P_app_ of silybin or LY.

**Figure 5 pharmaceutics-13-00628-f005:**
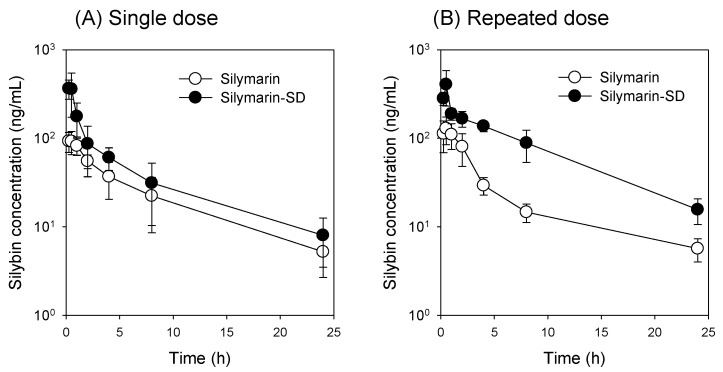
Plasma concentration–time profiles of silybin after (**A**) single and (**B**) repeated oral administration for five consecutive days of silymarin (20 mg/kg as silybin) and silymarin-SD (20 mg/kg as silybin) in rats with APAP-induced hepatotoxicity. Treatments followed the dosing schedule (Figure 1). Each data point represents the mean ± standard deviation of four rats per group.

**Figure 6 pharmaceutics-13-00628-f006:**
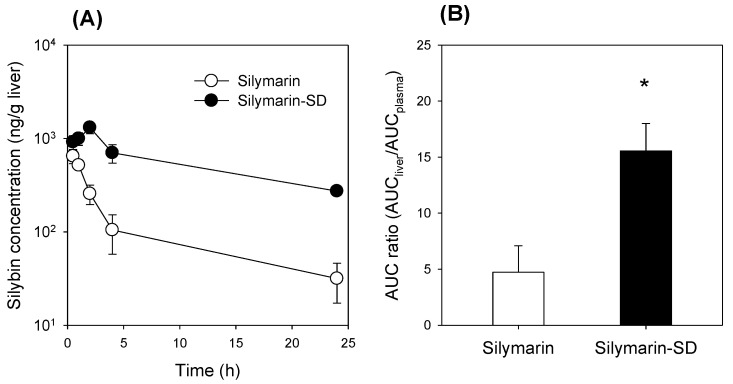
(**A**) Silybin concentrations in the liver. (**B**) Liver to plasma AUC ratio of silybin after the repeated oral administration for five consecutive days of silymarin (20 mg/kg as silybin) and silymarin-SD (20 mg/kg as silybin) in rats with APAP-induced hepatotoxicity. Treatment followed the dosing schedule in Figure 1. Each data represents mean ± standard deviation of four rats at individual time point per group. * *p* < 0.05, compared with silymarin group.

**Figure 7 pharmaceutics-13-00628-f007:**
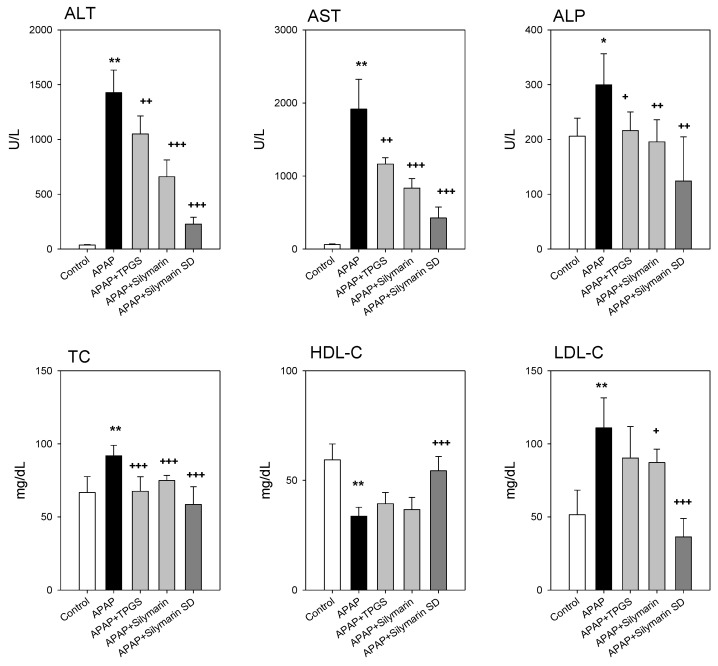
Alanine aminotransferase (ALT), aspartate aminotransferase (AST), alkaline phosphatase (ALP), total cholesterol (TC), high-density lipoprotein-cholesterol (HDL-C), and low-density lipoprotein-cholesterol levels (LDL-C) from control, acetaminophen (APAP)-induced hepatotoxic rat (APAP), APAP-induced hepatotoxic rats treated with TPGS 50 mg/kg (APAP+TPGS), silymarin (20 mg/kg as silybin) (APAP+Silymarin), and silymarin-SD (20 mg/kg as silybin) (APAP+Silymarin-SD). Treatment followed the dosing schedule in Figure 1. Each data represents mean ± standard deviation of four rats per group. * *p* < 0.05, and ** *p* < 0.01, compared with silymarin group; ^+^
*p* < 0.05, ^++^
*p* < 0.01, and ^+++^
*p* < 0.001, compared with APAP group.

**Figure 8 pharmaceutics-13-00628-f008:**
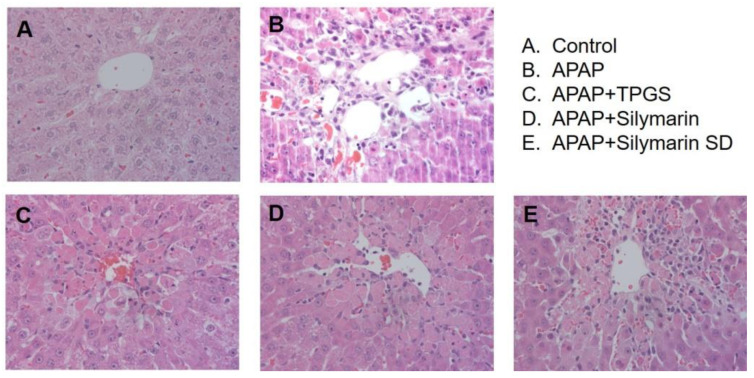
Representative liver histopathology hematoxylin and eosin (400×) images. The effect of silymarin-TPGS on APAP-induced hepatotoxicity in rats. (**A**) Control rats, (**B**) acetaminophen (APAP)-induced hepatotoxic rats, (**C**) APAP-induced hepatotoxic rats treated with TPGS (50 mg/kg) (APAP + TPGS), (**D**) APAP-induced hepatotoxic rats treated with silymarin (20 mg/kg as silybin) (APAP + silymarin), and (**E**) APAP-induced hepatotoxic rats treated with silymarin-SD (20 mg/kg as silybin) (APAP + silymarin-SD). Treatments followed the dosing schedule (Figure 1).

**Table 1 pharmaceutics-13-00628-t001:** Pharmacokinetic parameters of silybin.

Parameters	Single Administration		Repeated Administration	
	Silymarin	Silymarin-SD	Silymarin	Silymarin-SD
C_max_ (ng/mL)	106 ± 14.9	427 ± 147 *	146 ± 48.1	412 ± 168 *
T_max_ (h)	0.56 ± 0.31	0.40 ± 0.14	0.45 ± 0.11	0.44 ± 0.13
AUC_24h_ (ng∙h/mL)	578 ± 225	957 ± 350 *	547 ± 131	2040 ± 435 *^,+^
AUC_∞_ (n∙h/mL)	634 ± 239	1060 ± 406 *	587 ± 138	2190 ± 374 *^,+^
T_1/2_ (h)	6.77 ± 2.06	7.98 ± 2.80	6.74 ± 1.11	7.36 ± 1.32
MRT (h)	5.16 ± 0.47	7.66 ± 1.86 *	4.88 ± 0.72	8.32 ± 1.82 *

C_max_, maximum plasma concentration; T_max_: time to reach C_max_; AUC_24h_, area under the plasma concentration curve from zero to 24 h; AUC_∞_, area under the plasma concentration curve from zero to infinity; T_1/2_, elimination half-life; MRT, mean residence time. * *p* < 0.05, compared with silymarin itself, ^+^
*p* < 0.05, compared with a single administration.

## Data Availability

The data presented in this study are available upon request.

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
