# Peer review of "Enhanced Bioavailability and Efficacy of Silymarin Solid Dispersion in Rats with Acetaminophen-Induced Hepatotoxicity"

_pharmaceutics, 2021, doi:10.3390/pharmaceutics13050628_

Round 1

Reviewer 1 Report

The authors investigated bioavailability and efficacy of silymarin in rats with acetaminophen-induced hepatotoxicity by SD. The paper is in line with the journal scope. The following items are suggested for correction.

Line 53-55, rewrite the sentence

L 57-60 these information can be paced in the second paragrah. It is not necesarry to be an individual paragrah

The introdution need to be rewritten. Information on solubility enhancements of silymarin should be state in detail. What is the novlty of this study, which could not be seen from the introdution.

L80-86, please give the CAS nummber

L91-94 the X-ray diffraction section belongs to the characterization, not the preparation of silymarin-DS.

L103 What is silymarin SD? Is it the same as “silymarin-SD”?

L109  Does “silymarin-SD” refer to dired silymarin-SD? It is confused to name the solid dispersion. Because in section 2.2, “Silymarin-SD“ was in a solid state (frozen). “Silymarin-SD“was dired to produce dried “silymarin-SD”.

What is the load efficiency of silymarin in the SD?

Section 2.4 why the number of the control rats and APAP rats were 4? The number is too small to observe and obtain precise results.

Is there any relationship between the X-ray diffraction reults and the solubility of silymarin-SD?

Table 1 The significant digits of the data should be the same.

Reviewer 2 Report

The manuscript is well written. however, following suggestion can be considered.

  1. Please describe the detailed formulation development
  2. No results are provided for other API:Polymer ratio
  3. What is the glass transition temperature of the dispersion?
  4. Please include DSC data as it can give more information for miscibility of the API and polymer and stability of the dispersion.
  5. Is any supporting studies like FTIR is done to study the drug polymer interactions? If yes, it will be good to include.

Round 2

Reviewer 1 Report

The Authors have revised the manuscript thoroughly.